# Antibody Responses to SARS-CoV-2 and Common HCoVs in Hemodialysis Patients and Transplant Recipients: Data from the Dominican Republic

**DOI:** 10.3390/vaccines13090965

**Published:** 2025-09-11

**Authors:** Lisette Alcantara Sanchez, Eloy Alvarez Guerra, Dongmei Li, Samantha M. King, Shannon P. Hilchey, Qian Zhou, Stephen Dewhurst, Kevin Fiscella, Martin S. Zand

**Affiliations:** 1Clinical and Translational Science Institute, University of Rochester, Rochester, NY 14642, USA; lisette_alcantara@urmc.rochester.edu (L.A.S.); dongmei_li@urmc.rochester.edu (D.L.); 2Instituto Nacional de Coordinación de Trasplante, Santo Domingo 10103, Dominican Republic; eloydaniel.ag@gmail.com; 3Department of Medicine, Division of Nephrology, University of Rochester, Rochester, NY 14642, USA; samantha_king@urmc.rochester.edu (S.M.K.); shannon_hilchey@urmc.rochester.edu (S.P.H.); qian_zhou@urmc.rochester.edu (Q.Z.); 4Department Microbiology and Immunology, University of Rochester, Rochester, NY 14642, USA; stephen_dewhurst@urmc.rochester.edu; 5Department of Family Medicine, University of Rochester, Rochester, NY 14642, USA; kevin_fiscella@urmc.rochester.edu; 6Department of Public Health Sciences, University of Rochester, Rochester, NY 14642, USA

**Keywords:** COVID-19, SARS-CoV-2 vaccine, transplant recipients, hemodialysis, immunocompromised, common human coronavirus, cross-reactivity, antibody response, anti-S, anti-N

## Abstract

**Background:** Vaccination against SARS-CoV-2 has been pivotal in controlling the COVID-19 pandemic. However, understanding vaccine-induced immunity in immunocompromised individuals remains critical, particularly how prior exposure to other coronaviruses modulates immune responses. The influence of previous infections with endemic human coronaviruses (HCoVs), such as OC43, on SARS-CoV-2 immunity is not fully understood. This study evaluates antibody responses to COVID-19 vaccination in hemodialysis patients (HD), transplant recipients (TR), and healthy controls (CO), accounting for prior SARS-CoV-2 infection and baseline human coronavirus (HCoV) reactivity. **Methods:** We obtained longitudinal antibody measurements from 70 subjects (CO: *n* = 33; HD: *n* = 13; TR: *n* = 24) and assessed antibody kinetics across multiple post-vaccination time points using multivariate linear mixed modeling (MLMM). **Results:** Limited but measurable cross-reactivity was observed between SARS-CoV-2 and endemic HCoVs, particularly the β-coronavirus OC43. Pre-existing immunity in healthy individuals modestly enhanced vaccine-induced anti-spike (S) IgG responses, supported by post-vaccination increases in SARS-CoV-2 IgG. Prior SARS-CoV-2 infection significantly influenced anti-S and nucleocapsid (N) IgG responses but had limited impact on endemic HCoVs responses. Vaccine type and immune status significantly affected antibody kinetics. mRNA vaccination (BNT162b2) elicited stronger and more durable SARS-CoV-2 anti-S IgG responses than the inactivated CoronaVac vaccine, especially in immunocompetent individuals. Immunocompromised groups showed delayed or attenuated responses, with modest anti-S IgG cross-reactive boosting. Elevated anti-N IgG in CoronaVac recipients raised questions about its origin—infection or vaccine effects. MLMM identified key immunological and clinical predictors of antibody responses, emphasizing the critical role of host immune history. **Conclusions:** These findings highlight a constrained but meaningful role for HCoV cross-reactivity in SARS-CoV-2 immunity and vaccine responsiveness, underscore the need for infection markers unaffected by vaccination, and support development of broadly protective pan-coronavirus vaccines and tailored strategies for at-risk populations.

## 1. Introduction

The emergence of SARS-CoV-2 has highlighted the importance of understanding immune cross-reactivity with human coronaviruses (HCoVs), as it may impact disease progression, vaccine efficacy, and diagnostic accuracy [1,2,3]. Cross-reactivity occurs when an immune response, such as antibody production, targets similar antigens from different HCoV strains due to shared epitopes on surface proteins, potentially leading to diagnostic challenges or unintended immune effects [2,4,5]. SARS-CoV-2, a β-coronavirus, shares structural similarities with other HCoVs, particularly in its spike (S) and nucleocapsid (N) proteins, which can trigger cross-reactive immune responses [2,3,5,6]. Pre-existing immunity to seasonal α-(229E, NL63) and β-coronaviruses (OC43, HKU1) may alter immune responses to SARS-CoV-2 through antigenic cross-reactivity [2,4]. In immunocompromised individuals, such as hemodialysis patients (HD) and transplant recipients (TR), cross-reactivity may impair vaccine efficacy or viral clearance, increasing the risk of severe COVID-19 [7].

At the molecular level, SARS-CoV-2 and other HCoVs share key features, such as the receptor-binding domain (RBD) of the S1 subunit, which mediates viral entry via the angiotensin-converting enzyme 2 (ACE2) receptor, and serves as a primary vaccine target [8,9]. Cross-reactive antibodies from prior infections with endemic human coronaviruses (HCoVs) may provide partial immunity to SARS-CoV-2, potentially influencing disease severity, vaccine effectiveness, and immune responses [1,5,8,10,11]. However, the precise role of this cross-reactivity remains under investigation, including concerns regarding the theoretical risk of antibody-dependent enhancement (ADE) [5,8]. The widespread circulation of multiple HCoVs further complicates the assessment of past infections and the evaluation of vaccine-induced immunity, particularly in regions where these viruses co-exist [1,10,11]. Moreover, cross-reactivity poses significant challenges for serological diagnostics, as antibodies targeting shared epitopes may result in false-positive findings, thereby affecting clinical and epidemiological interpretations [2,3,5,12]. These challenges underscore the need for multiplex diagnostic assays and a clearer understanding of cross-reactive immunity to refine vaccination strategies and enhance public health interventions, especially for vulnerable populations [13,14,15]. Additionally, the impact of pre-existing immunity on disease severity appears to vary across populations, particularly among individuals with chronic kidney disease (CKD) and other comorbidities, thereby complicating clinical management [14,16].

Chronic kidney disease is a major health concern in the Dominican Republic, with 9358 active cases in 2023; among them, 5275 were on dialysis, including 4083 on hemodialysis [17]. Furthermore, approximately 500 individuals were on the kidney transplant waiting list, while 299 received a kidney transplant between 2019 and 2023 [18]. It is essential to gather data on the response to COVID-19 vaccines among individuals on hemodialysis (HD) and transplant recipients (TR) in the DR to better understand how factors such as demographics, environment, and prevalence of other common HCoV influence vaccine effectiveness, thereby improving healthcare outcomes locally and globally. The DR’s national vaccination campaign, launched in February 2021, prioritized high-risk groups, including those on HD and TR, due to their higher vulnerability to infection and severe outcomes [19]. Three vaccines were offered: BNT162b2 (Pfizer–BioNTech), AZD1222 (AstraZeneca–Oxford), and CoronaVac (Sinovac) [20]. By March 2024, 15 million doses had been administered to approximately 11 million individuals, with 68% having received at least one dose, and 56% fully vaccinated after two or more doses [21]. Despite this, there remains a lack of real-world or clinical trial data on vaccine responses in HD and TR patients in the DR. This is particularly important as local health and demographic factors, such as vaccine availability, access, and the ability to reach specific populations, can lead to significant variation in vaccine effectiveness.

Here, we report a secondary analysis of capillary blood samples that examines immune protection and cross-reactive immunity in high-risk populations by analyzing IgG responses to SARS-CoV-2 and four common circulating HCoVs (OC43, HKU1, 229E, NL63) in transplant recipients (TR) and hemodialysis patients (HD) in the Dominican Republic. As part of a study approved by Instituto Nacional de Coordinación de Trasplante (INCORT), we used a multiplex assay to simultaneously measure antibodies targeting the spike (S) and nucleocapsid (N) proteins of SARS-CoV-2 and the four common HCOV strains [5,8,22,23], while larger studies have examined antibody reactivity to SARS-CoV-2, no prior studies have looked at the correlation with antibodies directed against the common circulating coronaviruses in dialysis patients and renal transplant recipients, and none that we are aware of have comparison data for CoronaVac. This study also assessed potential cross-reactivity between SARS-CoV-2 and HCoVs, providing critical insights to refine diagnostic tools, inform vaccine strategies, and enhance clinical management of immunocompromised individuals.

## 2. Materials and Methods

### 2.1. Human Subjects Ethics Statement

This is the second report in a series of studies based on secondary analysis of capillary blood samples and de-identified data collected by INCORT in the Dominican Republic [23]. The original protocol was approved on 6 October 2020, by the National Health Bioethics Committee of the Dominican Republic (CONABIOS; INCORT.PI2020.004.01), with an amendment granting partial approval for our participation in 20 July 2021. The secondary analysis was subsequently approved in 12 January 2022 by the Institutional Review Board (IRB) at the University of Rochester Medical Center (IRB STUDY00006807). All research data were coded in accordance with the U.S. Department of Health and Human Services regulations for the protection of human subjects (45 CFR 46.101(b) (4)).

### 2.2. Study Design and Sample Collection

The methodological framework of the INCORT parent study was detailed in a prior publication [23]. The study enrolled 70 subjects from four hospitals affiliated with the national transplant recipient registry, as well as from dialysis units at three major national referral centers in the Dominican Republic. A control group was recruited from three COVID-19 vaccination clinics located in Santo Domingo. Recruitment and data collection were facilitated by medical staff through telephone calls, in-person interactions during hemodialysis sessions, and follow-up visits with transplant recipients.

The study population included three cohorts: transplant recipients (TR, *n* = 24), hemodialysis patients (HD, *n* = 13), and healthy controls (CO, *n* = 33), each further stratified by SARS-CoV-2 infection history (Figure 1). A total of 271 capillary blood samples were collected via volumetric absorptive micro-sampling (VAMS) prior to the administration of at least one dose of either the BNT162b2 (Pfizer-BioNTech) mRNA vaccine or the CoronaVac (Sinovac) inactivated SARS-CoV-2 vaccine. Sampling was conducted at five time points: baseline (Day 0, prior to the first vaccine dose), Day 21 or 28 (prior to the second dose, depending on the vaccine schedule), Day 42, and Day 70 (two and four weeks after the second dose), as well as prior to any booster dose, if administered, and six months after the final dose. The analysis included both homologous and heterologous booster regimens—homologous boosters involved the same vaccine as the primary series, while heterologous boosters used a different vaccine. For vaccine selection, subjects followed healthcare providers’ recommendations based on the national COVID-19 vaccination guidelines of the Dominican Republic [19]. All samples, initially stored by INCORT, were sent to the University of Rochester Medical Center for analysis of SARS-CoV-2 IgG-mediated immunity using the mPlex assay [5,8,22,23].

The inclusion criteria of the parent study involved adults aged 18 or older who were either receiving hemodialysis or had undergone solid organ transplantation and were willing to receive a COVID-19 vaccine. To facilitate meaningful comparison with these immunocompromised cohorts, a control group was selected based on the absence of common medical conditions (i.e., hypertension, diabetes, cardiovascular disease, chronic respiratory conditions, immunodeficiency, and both degenerative and autoimmune diseases) that are known to elevate the risk of SARS-CoV-2 infection and severe outcomes [24]. Our sub-study excluded individuals who tested positive for COVID-19 via PCR at enrollment or had previously received any COVID-19 vaccine doses. Focusing on unvaccinated subjects enabled a longitudinal analysis of immune responses to subsequent COVID-19 vaccinations and infections from both SARS-CoV-2 and human coronaviruses (HCoVs), as well as key variables related to dialysis, organ transplantation, and COVID-19 vaccination status. As reported in the initial paper of this study series [23], the clinical variables are summarized in Appendix A (reproduced from that source).

Our analysis included data previously collected by INCORT through face-to-face interviews conducted by INCORT research personnel. These interviews included a self-reported questionnaire that gathered information on demographics, health status, and COVID-19 infection and vaccination history. Data compiled age, sex, race, ethnicity, highest level of education, and the number of vaccine doses received. This comprehensive set of variables was used in a Multivariate Linear Modeling to analyze the complex relationships between condition-specific factors, COVID-19 infection, vaccination status, and other health-related variables within the study population. Cohort demographics, COVID-19 infection, and vaccination status are summarized Table 1.

### 2.3. mPlex-CoV Assay

All samples were collected as part of the parent study and processed in as described previously [8,22,23]. Antibodies were eluted from VAMS swabs by immersing them in 200 μL of extraction buffer (0.5% Tween 20, 1% BSA in PBS) at 4 °C with gentle agitation overnight. The eluents were aliquoted into 100 μL tubes and stored at 4 °C prior to multiplex assay analysis [6]. Quantified sample IgG concentrations targeting the SARS-CoV-2 spike (S) and nucleocapsid (N) protein antigens from various viral variants, including HCoVs (OC43, HKU1, 229E, NL63) as previously described [5,8,22]. Assay targets included recombinant, trimerized SARS-CoV-2 S- and N-proteins, as well as recombinant S1, S2, and RBD proteins (SinoBio, Beijing, China). Recombinant proteins were conjugated to magnetic microsphere beads (Luminex, Austin, TX, USA) at 40 pmole/106 beads as previously described. Eluent from VAMS devices (200 μL) for the IgG assay was diluted 1:1000. A total of 50 μL of the diluted sample was combined with 50 μL of the mPlex-CoV bead panel suspension and added to wells of 96-well plates in duplicate (Microplate, GBO, Kremsmünster, Austria). After washing, the bound IgG was detected with phycoerythrin-conjugated anti-human IgG (Southern Biotech, Birmingham, AL, USA, Cal No: 2040-09, 2050-09), and the median fluorescence intensity (MFI) was measured using a Luminex MagPix (R&D Systems, Inc., Minneapolis, MN, USA) [8,22].

### 2.4. Measurement of Hemoglobin (Hgb) and Adjustment of Antibody Concentration

We used hemoglobin (Hgb) values measured in the VAMS eluent to adjust capillary blood IgG concentrations for plasma volume, following published procedures [6,23]. From Hgb concentration, we calculated hematocrit and used it to refine IgG values, which allowed us to estimate serum IgG concentrations for SARS-CoV-2 and the seasonal coronaviruses OC43, HKU1, 229E, and NL63. To determine anti-S and anti-N IgG antibody concentrations, we applied standard curves generated from pooled positive sera controls, as described previously [8,22].

### 2.5. Multidimensional Scaling (MDS) and Multivariate Linear Mixed Model (MLMM) Statistical Analysis

The demographics, subject characteristics, vaccination history details, and the status of the COVID-19 infection are summarized using frequency distributions and summary statistics. We evaluated the differences between study groups using chi-square or Fisher’s exact tests for categorical variables (Table 1).

Multidimensional Scaling (MDS) analysis of anti-S and anti-N IgG for common HCoVs—OC43, HKU1, 229E, NL63 was used to examine their association with anti-S and anti-N IgG SARS-CoV-2, identify groups, and reduce complexity. Using pairwise distances among HCoV-specific IgGs, MDS was applied to produce a set of dimensions that capture the main patterns in these variables. The first few MDS-derived dimensions, which accounted for the most variance, were selected as composite predictors and incorporated into the MLMM [25].

The MLMM was then fitted with these MDS-derived dimensions as predictors of the outcomes (anti-S and anti-N IgG SARS-CoV-2 IgG), effectively reducing multicollinearity and improving interpretability. The model was specified with appropriate fixed effects (e.g., the MDS dimensions) and random effects (e.g., grouping by cohort, prior COVID-19 infection, vaccine type), preserving the hierarchical structure of the data. The MLMM was then estimated with the coefficients for the MDS-derived predictors, assessing the relationships with SARS-CoV-2 anti-S IgG concentrations while accounting for random effects. MDS visualizations highlighted study groups and trends in anti-S IgG that aligned with the relationships captured by the MLMM coefficients. This combined approach facilitated dimensionality reduction, improved model stability, and clarified key patterns between SARS-CoV-2 anti-S IgG levels and anti-S IgG for common HCoVs. By leveraging both statistical inference (via MLMMs) and exploratory visualization (via MDS), we gained robust insights into our complex multivariate dataset.

Statistical analysis was performed using R version 3.5.1 (R Core Team, 2017). The significance level for all tests was set at *p* < 0.05.

## 3. Results

### 3.1. Participant Characteristics, Vaccination Status, and COVID-19 History

As reported in the first paper of this series, the population included 70 subjects across three cohorts: Control (CO, *n* = 33), Hemodialysis (HD, *n* = 13), and Transplant Recipients (TR, *n* = 24) [23]. We found no statistically significant differences in age or gender among cohorts (*p* = 0.150), although males comprised a higher proportion of the HD and TR groups (57%) (Table 1). Most subjects self-identified as Black and Hispanic or Latino (*p* < 0.001), reflecting Dominican Republic demographics, where 74% of the population self-identifies as mixed race [26,27]. We observed significantly higher educational attainment in the TR cohort (*p* < 0.001). Overall, 44% of subjects reported prior COVID-19 infection or a positive test result before vaccination. Recruitment imbalances across cohorts were primarily driven by early vaccination campaigns that prioritized high-risk populations, such as individuals on dialysis or post-transplant [19]. As a result, INCORT reported that enrolling unvaccinated participants had become increasingly difficult by the time the study began.

There were significant differences in SARS-CoV-2 vaccination status across cohorts (*p* < 0.001), with the TR group showing the highest rate of full vaccination with one booster dose (42%), while vaccine type and booster regimen distribution varied, these differences were not statistically significant (*p* = 0.066). Among subjects who received heterologous boosters, the most common sequence was an initial CoronaVac dose followed by a BNT162b2 booster. In contrast, homologous vaccination primarily involved two doses of BNT162b2 (87.5%), with only one participant receiving two doses of CoronaVac. Previous COVID-19 infection was reported more frequently in the TR group (63%) compared to the HD (46%) and CO (30%) groups, with these differences approaching statistical significance (*p* = 0.053). In contrast, there were no statistically significant differences in COVID-19 positivity during the study period, as only 21.43% of participants reported being aware of a positive test prior to enrollment.

### 3.2. Co-Factor Analysis of Antibody Responses to SARS-CoV-2 and Common Human Coronaviruses (HCoVs)

We applied Multivariate Linear Modeling (MLMM) to examine the influence of multiple co-factors on IgG antibody responses to SARS-CoV-2 and common human coronaviruses (HCoVs), including OC43, HKU1, 229E, and NL63 (Figure 2). This analysis enabled the evaluation of the relative contributions of variables, including COVID-19 vaccination history, prior SARS-CoV-2 infection, and immunocompromised status, to the development of antibody levels before and after vaccination. Among these factors, prior SARS-CoV-2 infection and vaccine type emerged as the strongest predictors of elevated anti-Spike IgG levels (left panel).

A history of self-reported COVID-19 positivity prior to vaccination was significantly associated with increased SARS-CoV-2 anti-S IgG responses (*p* = 0.001). Additionally, a highly significant interaction between cohort and prior SARS-CoV-2 infection (*p* < 0.001) indicated that individuals with hybrid immunity, defined as natural infection followed by vaccination, had the highest antibody levels (Figure 2, left panel). Vaccine type also influenced responses; CoronaVac priming was associated with a modest but significant increase in anti-S IgG levels (*p* = 0.021), though its effect on cross-reactive responses was limited. In relation to cross-reactivity with common circulating HCoVs, prior SARS-CoV-2 infection was associated with significantly higher anti-S IgG levels against OC43 (*p* = 0.004) and HKU1 (*p* = 0.012), also β-corona virus strains, suggesting cross-reactive immune responses.

Findings supporting cross-reactive responses were further corroborated by correlation analyses (Appendix A), which demonstrated significant positive associations between SARS-CoV-2 anti-S IgG antibodies and those targeting common HCoVs. Moderate correlations were observed between SARS-CoV-2 anti-S IgG and the β-coronaviruses OC43 (R=0.477) and HKU1 (R=0.541), while weaker correlations were noted with the α-coronaviruses 229E (R=0.295) and NL63 (R=0.353). The strongest correlation was detected between the β-HCoVs OC43 and HKU1 (R=0.877, *p* < 0.001). Furthermore, stratification by cohort revealed stronger inter-virus correlations in the HD and TR groups compared to CO, particularly between SARS-CoV-2 and OC43 (HD: R=0.665; TR: R=0.596). These findings suggest that altered immune status may facilitate broader or more interconnected antibody responses, consistent with the enhanced cross-reactivity identified in the MLMM [2,10,12,16].

For anti-N IgG responses (Figure 2, right panel), MLMM analysis identified prior SARS-CoV-2 infection as the primary factor associated with elevated antibody levels (*p* < 0.001). As expected, individuals with reported prior SARS-CoV-2 infection exhibited markedly higher SARS-CoV-2 anti-N IgG responses, along with increased anti-N IgG against common HCoVs, particularly HKU1 (*p* = 0.012), OC43 (*p* = 0.004), and to a lesser extent 229E (*p* = 0.013), suggesting potential cross-reactivity. In contrast, those without prior SARS-CoV-2 infection showed minimal anti-N IgG reactivity against HCoVs, consistent with limited nucleocapsid exposure. Additionally, similar to anti-S responses, the TR group demonstrated significantly reduced HCoVs anti-N IgG levels, underscoring the suppressive impact of immunosuppression on humoral immunity targeting nucleocapsid proteins.

Correlation analyses also supported these observations (Appendix A), revealing moderate positive corlletions between SARS-CoV-2 anti-N IgG and antibody levels against common HCoVs: HKU1 (R=0.543), 229E (R=0.589), and OC43 (R=0.456). Strong correlations were most pronounced in the HD cohort, with values reaching R=0.708 for SARS-CoV-2 vs. HKU1 and R=0.845 for OC43 vs. HKU1, suggesting heightened immune stimulation or broader cross-reactivity in this group. The TR cohort displayed moderate correlations, whereas controls exhibited weaker associations and generally lower antibody levels. Strong correlations between OC43 and HKU1 across all cohorts likely reflect their genetic similarity.

### 3.3. Antibody Responses to SARS-CoV-2 and Common Human Coronaviruses in Hemodialysis, Transplant, and Control Groups

To examine SARS-CoV-2 vaccine-induced immune responses, potential cross-reactive effects with common human coronaviruses (HCoVs), and differences across three cohorts, we conducted longitudinal measurements of IgG concentrations targeting HCoV spike and nucleocapsid proteins in HD, TR, and CO (Figure 3). We assessed changes in anti-S and anti-N IgG levels in relation to both SARS-CoV-2 and HCoVs responses. Virus-specific IgG profiles are presented in (Appendix A). We hypothesized that higher HCoVs IgG levels correlate with stronger SARS-CoV-2 anti-S IgG responses post-vaccination and booster across all groups, with differences between HD, TR, and controls reflecting the extent and nature of immunocompromise, potentially affecting cross-reactivity. This analysis allowed us to evaluate how pre-existing B cell immunity to HCoVs may modulate humoral responses to COVID-19 vaccination and boosting.

At baseline, CO showed the highest anti-S IgG concentrations, particularly for SARS-CoV-2 (*p* < 0.001), followed by OC43 (*p* = 0.022) and HKU1 (*p* < 0.001), reflecting likely prior exposure (Figure 3A). Following vaccination, SARS-CoV-2 anti-S IgG levels rose significantly over time across all three cohorts (*p* < 0.001), with a markedly greater increase in individuals with prior SARS-CoV-2 infection (*p* < 0.001), consistent with a strong secondary immune response, as detailed in Figure 4. A significant interaction between COVID-19 infection history and vaccine type (*p* = 0.030) indicates that prior exposure modulates vaccine-induced immunity in a platform-dependent manner. These effects were more pronounced and sustained in recipients of BNT162b2 compared to CoronaVac, particularly within the CO and HD groups, as detailed in Figure 6. Among cohorts, the CO group exhibited a modest but consistent increase in SARS-CoV-2 anti-S IgG over time (*p* = 0.012), with levels remaining significantly higher than those targeting HCoVs anti-S (*p* < 0.001). In the HD group, we observed a delayed and significant increase in anti-S IgG (*p* = 0.028). In contrast, the TR group showed modest and statistically significant responses across all time points (*p* < 0.001), consistent with immunosuppression and B cell dysfunction.

SARS-CoV-2 anti-N IgG levels increased progressively after the second vaccine dose and booster—particularly, in the HD cohort (*p* = 0.016). In contrast, baseline anti-N IgG levels for OC43 (*p* < 0.001) and NL63 (*p* < 0.001) were significantly higher in the CO and TR cohorts compared to HD, indicating pre-existing immunity. However, no significant changes anti-N IgG levels were observed over time across cohorts (*p* = 0.110), suggesting potential differences in prior HCoV exposure among the groups.

### 3.4. Anti-Spike and Anti-Nucleocapsid IgG Antibody Responses to SARS-CoV-2 and Common Human Coronaviruses in Previously Infected and Non-Infected Subjects

We evaluated the impact of prior SARS-CoV-2 infection and pre-existing immunity to HCoVs on vaccine-induced immune responses (Figure 4). Anti-S IgG antibodies were measured at multiple time points and stratified by COVID-19 history and cohort. As hypothesized, prior SARS-CoV-2 infection was significantly associated with higher overall anti-S IgG levels following vaccination (*p* < 0.001) (Figure 4B). At the baseline (day 0), individuals with a history of SARS-CoV-2 infection exhibited significantly elevated responses to both SARS-CoV-2 anti-S and anti-N IgG compared to infection-naive subjects (anti-S: *p* < 0.001; anti-N: *p* = 0.003), consistent with residual immunity from natural infection.

Following vaccination, a modest but statistically significant increase in OC43 anti-S IgG levels was observed over time (*p* = 0.025), particularly among individuals who received two or more vaccine doses (*p* = 0.017), suggesting potential cross-reactive boosting of anti-S IgG. Similarly, prior SARS-CoV-2 infection was associated with small but statistically significant increase antibody levels for 229E (*p* = 0.026), but not HKU1 (*p* = 0.230). For NL63, no significant differences were observed between individuals with and without prior infection, as responses remained low and stable across all three groups and timepoints in both cases. The CO group (*p* = 0.095), HD group (*p* = 0.026), and TR group (*p* = 0.011) exhibited similar overall trends.

For anti-N IgG responses, as anticipated, prior SARS-CoV-2 infection was the dominant predictor of elevated antibody levels at baseline (Figure 5B). Individuals with prior infection exhibited markedly higher anti-N IgG levels against SARS-CoV-2 (*p* < 0.001), along with evidence of cross-reactivity to OC43 (*p* = 0.004), NL63 (*p* < 0.001), and 229E (*p* = 0.048). This pattern was not observed for HKU1 (*p* = 0.012), which consistently exhibited low levels over time. Conversely, individuals without prior infection showed minimal anti-N IgG reactivity, consistent with the absence of nucleocapsid antigen exposure through vaccination alone.

To assess the durability of the anti-N IgG response, we evaluated the interaction between time and prior infection status. The interaction term was negative and statistically significant for SARS-CoV-2 anti-N IgG antibodies (*p*) = 0.001), indicating a modest but significant decline in antibody levels over time among individuals with prior infection. Similarly, effect estimates over time for OC43 and NL43 (*p* = 0.510; *p* = 0.720, respectively) indicate these initially increased levels were not consistent across all time points, suggesting that COVID-19 infection does not broadly induce cross-reactive boosting of anti-N IgG responses.

Collectively, these findings show that prior SARS-CoV-2 infection strongly influences initial anti-N IgG responses to SARS-CoV-2 and HCoVs post-vaccination, but has limited impact on their persistence over time.

### 3.5. Anti-Spike and Anti-Nucleocapsid IgG Antibody Responses to SARS-CoV-2 and Common Human Coronaviruses and Vaccine Type

We examined the influence of vaccine type on antibody responses among different patient cohorts, specifically comparing CoronaVac (inactivated virus vaccine) to BNT162b2 (mRNA vaccine) recipients across HD, TR, and CO. Our MLMM analysis revealed that vaccine type was significantly associated with antibody responses to several viral antigens (Figure 2). Significant effects were observed for anti-S IgG responses to SARS-CoV-2 (*p* = 0.021), although the interaction between cohort and vaccine type did not achieve significance (*p* = 0.083) (Figure 6). Similarly, days after the first dose and prior SARS-CoV-2 infection significantly influenced anti-S IgG response (*p* < 0.001). Vaccine dose number was not a significant independent predictor in the adjusted models (*p* = 0.245).

Vaccine type-specific effects were particularly evident in individuals primed with BNT162b2 vaccine, who exhibited significantly higher SARS-CoV-2 anti-S IgG levels compared to those who received CoronaVac, especially within the CO group (estimate = −0.294, *p* = 0.003). Among HD, CoronaVac priming was associated with both lower peak antibody levels and a steeper decline over time (estimate = −0.122, *p* = 0.041). The most impaired response was observed in the TR group, where CoronaVac-primed individuals showed markedly reduced anti-S IgG levels (estimate = −0.629, *p* = 0.026), consistent with the effects of chronic immunosuppression.

Regarding cross-reactivity with HCoVs anti-S IgG responses, CoronaVac was associated with a significant increase in OC43 anti-S IgG levels (*p* = 0.079) and HKU1 anti-S IgG levels (*p* = 0.011). Cohort and vaccine type-interactions were also observed for OC43 (*p* = 0.048), with higher antibody responses in HD individuals vaccinated with CoronaVac. Additionally, for 229E anti-S IgG, vaccine type had a significant effect (*p* = 0.045) and a strong cohort by vaccine-type interaction (*p* < 0.001), with CoronaVac vaccination showing reduced responses in TR (estimate = −0.452, *p* = 0.016). For anti-S IgG NL63, vaccine type was significant (*p* = 0.037); however, no notable interaction effects were detected, and antibody levels remained consistently low over time. Moreover, prior SARS-CoV-2 infection was linked with higher IgG responses to OC43 (*p* = 0.004) and HKU1 (*p* = 0.012), particularly in BNT162b2-vaccinated HD. CoronaVac-primed TR showed significantly diminished responses to HKU1 (*p* = 0.014) and 229E (*p* = 0.048), highlighting compromised breadth under immunosuppression.

These findings were reinforced by correlation analysis (Appendix A). The strongest inter-virus IgG correlations were observed between OC43 and HKU1 (R=0.877, (*p* < 0.001). Anti-S IgG responses to SARS-CoV-2 were also positively correlated with OC43 in HD (R=0.665) and TR (R=0.596), while CO showed much weaker associations with 229E (R=0.295). These patterns suggest that altered immune status in HD and TR cohorts leads to more interconnected or broadly reactive humoral responses.

Our analysis of anti-N IgG responses revealed that vaccine type significantly modulates the magnitude and kinetics of antibody production, with notable differences between CoronaVac and BNT162b2 recipients across cohorts (Figure 7). A highly significant interaction between cohort and vaccine type (*p* < 0.001) suggests that the influence of vaccine type on SARS-CoV-2 anti-N IgG responses varies substantially between groups.

CoronaVac recipients in the CO group showed elevated anti-N IgG to HKU1 (estimate = 0.162, *p* = 0.021), that suggests cross-reactive responses, likely due to the whole-virus nature of the vaccine. Similarly, in HD with hybrid immunity due to prior SARS-CoV-2 infection, CoronaVac further amplified anti-N IgG responses OC43 (*p* = 0.067) and HKU1 (*p* = 0.015), also suggesting cross-reactive response, while BNT162b2 vaccine, which lack N antigen, induced limited cross-reactivity. TR again showed reduced anti-N IgG responses overall, particularly to SARS-CoV-2 and HKU1 (estimate =−0.237, *p* = 0.014).

Correlation analysis (Appendix A) revealed moderate interrelatedness among anti-N IgG responses. SARS-CoV-2 anti-N IgG correlated positively with HKU1 (R=0.543), 229E (R=0.589), and OC43 (R=0.456). These associations were most pronounced in the HD cohort, where correlations peaked at R=0.708 (SARS-CoV-2 vs. HKU1) and R=0.845 (OC43 vs. HKU1), suggesting broader immune activation. TR displayed moderate correlations, while controls exhibited weaker associations.

Collectively, these findings indicate that vaccine type, prior exposure, and cohort immune competence influence not only the magnitude of anti-N IgG at baseline and post-vaccination but also the extent of cross-reactive responses, particularly among individuals with altered immune status. The data highlight the importance of cumulative nucleopcasid antigen exposure in shaping the humoral landscape.

## 4. Discussion

Clinical and immunological factors influence both the magnitude and cross-reactivity of IgG antibody responses to SARS-CoV-2 and related common human coronaviruses (HCoVs) [15,28,29,30,31], particularly in immunocompromised populations such as hemodialysis (HD) patients and transplant recipients (TR) [32,33]. Using multivariate linear modeling (MLMM), we systematically evaluated how pre-existing immunity to SARS-CoV-2 and HCoVs, vaccination status, and immune competence shape humoral responses to COVID-19 vaccination and boosting (Figure 2). By examining antibody responses against spike and nucleocapsid proteins across multiple coronaviruses, we identified prior SARS-CoV-2 infection and vaccine type as the strongest predictors of broadly reactive IgG responses, while also revealing a complex interplay with cohort-specific factors in shaping spike-specific immunity. emphasizing the role of hybrid immunity in enhancing responses to both SARS-CoV-2 and endemic HCoVs, aligning with previous reports [12,34].

Hybrid immunity and repeated vaccination seems to enhanced IgG responses to SARS-CoV-2 and endemic HCoVs, though dose number was not an independent predictor after adjustment. Common HCoVs triggered some cross-reactive anti-N IgG, with modest gains in low responders using less immunogenic platforms. Prior exposure to β-coronaviruses (e.g., OC43) boosted anti-S IgG post-vaccination—likely via memory B cells targeting conserved spike epitopes—especially in healthy individuals [10,16,35,36]. This effect was weaker in immunocompromised groups (HD, TR), and in some cases, absence of prior HCoVs immunity appeared to enhance de novo SARS-CoV-2 responses, reflecting complex immune imprinting dynamics [30,37].

Post-vaccination, anti-SARS-CoV-2 IgG increased in all groups—strongest in healthy individuals, delayed in HD, and reduced in TR—reflecting effects of uremia, inflammation, and immunosuppression [1,33,38,39,40]. Cross-reactive anti-S IgG to HCoVs remained mostly stable, with slight increases in HD and minimal change in TR (Figure 3A). A significant OC43 anti-S IgG rise in healthy individuals likely stemmed from conserved S2-domain epitopes [16,35,41], with little nucleocapsid cross-reactivity, while cross-reactivity may aid protection or response speed [30], it could also misdirect immunity toward non-neutralizing targets, consistent with original antigenic sin [35,37].

Immunosuppressive therapies, including mTOR inhibitors, calcineurin inhibitors, and corticosteroids, markedly impair humoral responses to SARS-CoV-2 infection and vaccination [39,42,43,44]. In the TR group, most subjects were more than 25 months post-transplant (66.7%) and maintained on potent immunosuppressants, including Everolimus (70.8%) and Cyclosporine (79.2%) (Appendix A). Despite these therapies, longer transplant-to-vaccination intervals were associated with delayed but detectable antibody responses, suggesting partial immune recovery over time [43,44]. Nevertheless, antibody titers and neutralizing capacity remained substantially lower than in immunocompetent individuals [43,44].

In HD patients given BNT162b2, prior SARS-CoV-2 infection boosted anti-S IgG to OC43 and HKU1, aligning with past studies [34,35]. TR vaccinated with CoronaVac showed weaker responses to 229E and HKU1, suggesting vaccine type and immunosuppression limit cross-reactivity [45]. Correlations between OC43 and HKU1 IgG in HD and TR may reflect altered immunity or repeated exposures [41,46]. BNT162b2 consistently elicited higher and durable responses across groups (Figure 6), confirming the stronger immunogenicity of mRNA vaccines compared to CoronaVac [9,29,42,47,48]. Although dose number was not an independent predictor, extra doses may benefit low responders using weaker platforms [49]. These findings support tailored strategies for immunocompromised patients, including personalized dosing and more immunogenic or adjuvanted vaccines [40,42,49,50,51].

Anti-N IgG responses were influenced by vaccine platform, infection history, and immune status (Figure 2). CoronaVac recipients, particularly those previously infected, displayed elevated anti-N IgG against HCoVs like HKU1 and OC43, consistent with the broader antigenic profile of whole-virus vaccines [48,52] (Figure 7). In contrast, BNT162b2, which lacks the nucleocapsid antigen, elicited minimal anti-N IgG responses [29,52]. Prior infection was the most significant driver of elevated anti-N IgG and cross-reactivity, reinforcing the importance of natural infection in broadening humoral responses [16,35,52]. Interestingly, TR subjects did not show markedly reduced anti-N IgG, while HD individuals with hybrid immunity demonstrated robust cross-reactivity, suggesting some compensatory mechanisms despite immune dysfunction [47]. Correlations between SARS-CoV-2 and HCoVs anti-N IgG responses, especially in HD, suggest cross-recognition of conserved epitopes [41,46].

Fluctuations in anti-N IgG levels may reflect reinfection or “back-boosting,” whereby prior exposure enhances existing immunity (Figure 5B). These changes, observed alongside variations in anti-S IgG, likely reflect vaccine-induced immune recall rather than new infections [30,31,53]. This phenomenon was particularly notable among recipients of CoronaVac, a whole-virus vaccine that induces both anti-S and anti-N responses [9,54,55,56]. These findings are consistent with reports of back-boosting following vaccination or heterologous exposure [29,30]. However, the rapid decline in anti-N IgG levels among previously infected individuals, especially in TR, limits its utility as a reliable marker of prior infection in immunosuppressed patients, and the presence of N-protein in the CornaVac whole virus vaccine limits the utility of anti-N IgG in identifying infection up to 4–6 weeks post-vaccination [30].

The study population showed no significant differences in age or gender across cohorts, although a higher proportion of males was observed in the HD and TR groups (Table 1). The predominance of Black and Hispanic or Latino subjects is consistent with the demographic trends in the Dominican Republic, where race and ethnicity are categorized distinctly. The TR cohort demonstrated higher levels of educational attainment and vaccination coverage, reflecting the success of targeted public health interventions, while variations in prior SARS-CoV-2 infection rates and vaccine regimens were noted, these differences did not reach statistical significance, suggesting overall consistency in exposure and immunization practices across the cohorts.

Our study has several limitations, most notably the small sample sizes. The rapid vaccine rollout in the Dominican Republic presented challenges for INCORT in recruiting unvaccinated individuals, particularly among immunocompromised groups. Vaccine hesitancy among hemodialysis patients—possibly related to poor health—further reduced participation. In the CO group, lower vaccination rates likely reflect hesitancy among younger, healthier individuals and fewer interactions with healthcare services [57]. In contrast, delays in the HD group may be attributed to misinformation or inadequate provider-patient communication [58]. Nonetheless, recommendations from nephrologists have been shown to improve vaccine uptake [58]. In contrast, 92% of TR received two or more doses, likely due to higher education levels, clinical vulnerability, and greater health engagement [33,59]. Age differences between groups limited control for confounding factors, making it difficult to separate age effects from group-specific effects. Future work should include a larger sample size in the three parallel cohorts. Extended follow-up periods are also necessary to evaluate long-term immune responses and to examine the effects of different vaccination schedules. Despite these challenges, the study population was demographically diverse in age, sex, and ethnicity, reflecting the broader national context.

In summary, vaccine platform, immune status, and prior SARS-CoV-2 exposure significantly shape the magnitude and cross-reactivity of humoral responses. mRNA vaccines provided the most robust and durable immunity, especially in vulnerable groups [29]. Our findings support incorporating non-spike antigens—such as nucleocapsid and conserved S2 domains—into future vaccine designs to broaden protection against variants and related coronaviruses [30,34,45,52,53]. These results have key implications for vaccine policy and immune monitoring. Immune responses vary widely by host factors and prior HCoVs exposure, with OC43 priming spike-specific responses in healthy individuals. In contrast, immunocompromised groups (HD, TR) show weaker responses, highlighting the need for tailored strategies, including additional doses or adjuvanted vaccines. Modest HCoV antibody boosts post-vaccination suggest cross-reactivity should inform both surveillance efforts and the design of pan-coronavirus vaccines.

## 5. Conclusions

This study provides evidence for limited but measurable cross-reactivity between SARS-CoV-2 and endemic human coronaviruses (HCoVs), particularly β-coronavirus OC43. Cross-reactive anti-S IgG responses were most evident in healthy individuals with prior HCoVs exposure, suggesting that pre-existing immunity may modestly enhance vaccine-induced responses through immunological priming. Post-vaccination increases in HCoV-specific antibodies, while modest, further support the presence of antigenic overlap, with implications for immune surveillance and serological interpretation. Reactivity and fluctuations in anti-N IgG responses—especially after CoronaVac, which induces and boosts these antibodies—highlight the need for alternative markers of infection that are unaffected by vaccination. By integrating multi-level mixed modeling (MLMM), this study identified key immunological and clinical co-factors that shape antibody responses, reinforcing the relevance of host immune history in vaccine response profiling. These findings emphasize the potential, yet constrained, role of HCoVs cross-reactivity in SARS-CoV-2 immunity and support the rationale for developing broadly protective, pan-coronavirus vaccine platforms.

## Figures and Tables

**Figure 1 vaccines-13-00965-f001:**
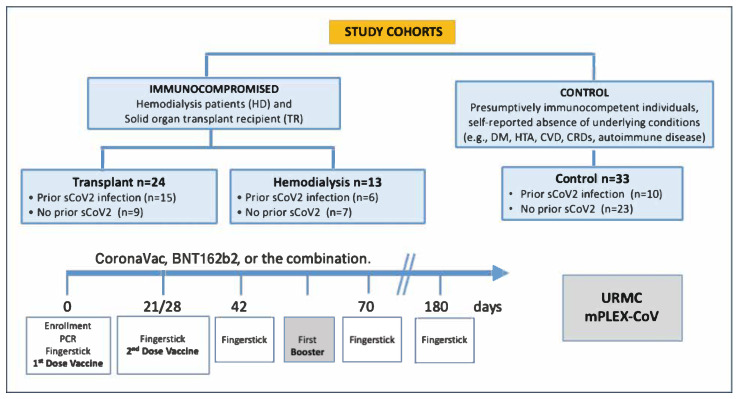
Data flow diagram illustrating study design and cohorts. Reproduced from [23]. A total of 271 capillary blood samples were collected from 70 participants across five time points post-vaccination. Subjects were grouped into control (*n* = 33), hemodialysis (*n* = 13), and transplant (*n* = 24) cohorts, stratified by prior COVID-19 infection. Samples were collected at day 0 (pre-first dose), day 21/28 (pre-second dose), days 42 and 70 (2 and 4 weeks post-second dose), before any booster if applicable, and 6 months after the last dose.

**Figure 2 vaccines-13-00965-f002:**
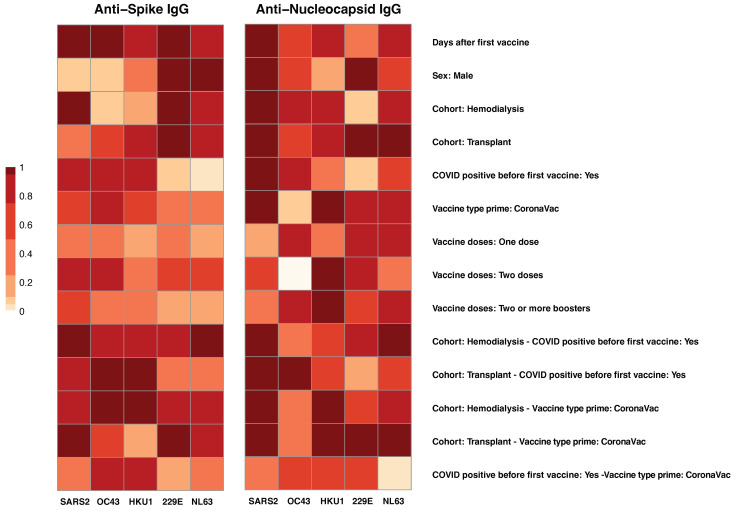
Factors Associated with anti-spike and anti-nucleocapsid IgG responses. The heatmap visualization shows the results of Multivariate Linear Modeling (MLMM) analysis of IgG responses to spike (**left panel**) and nucleocapsid (**right panel**) antigens from SARS-CoV-2 (SARS2) and four common human coronaviruses (HCoVs): OC43, HKU1, 229E, and NL63. Each row represents a co-factor, (e.g., vaccine history, prior SARS-CoV-2 infection, clinical cohort, immunocompromised status), while each column corresponds to one of the combination of viral strain and surface antigens (S, N). Color intensity, scaled from 0 (light) to 1 (dark red), reflects the relative strength of association for each factor with the corresponding antibody response. MLMM was used to identify key predictors of antibody responses and to adjust for potential confounding factors.

**Figure 3 vaccines-13-00965-f003:**
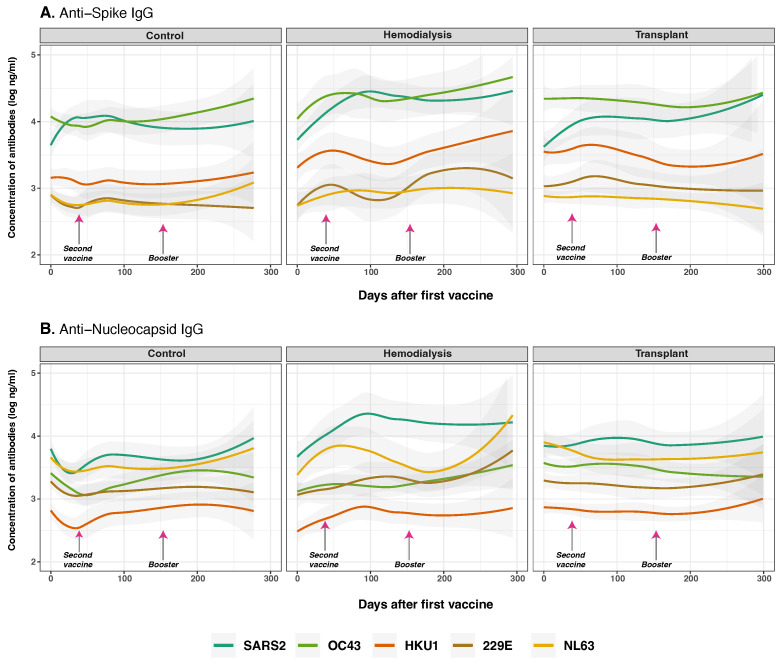
Anti-spike and anti-nucleocapsid IgG antibodies for SARS-CoV-2 and common human coronaviruses. Concentrations of total anti-S (panel (**A**)) and anti-N (panel (**B**)) IgG for SARS-CoV-2 (dark green lines), OC43 (olive green lines), HKU1 (orange lines), 229E (brown lines), NL63 (golden lines) measured in control (*n* = 33), hemodialysis (*n* = 13), and transplant recipient (*n* = 24) cohorts. Day 0 corresponds to the first vaccination. Arrows show the mean days for the second (36.6 ± 34.1 days) and booster (154.3 ± 41.1 days) doses. Individual data points were pooled and fitted with spline curves for each cohort, and gray bands indicate 95% confidence intervals.

**Figure 4 vaccines-13-00965-f004:**
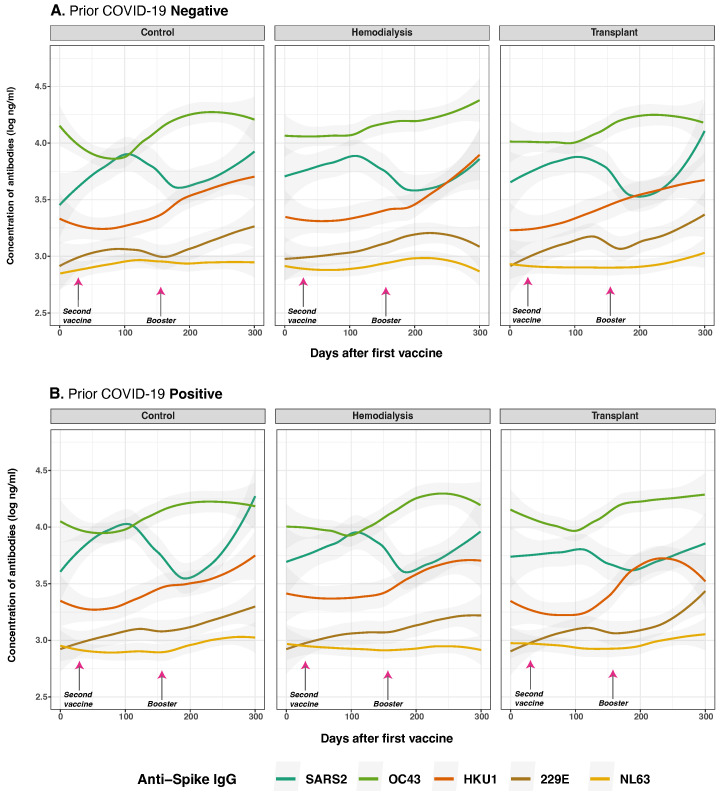
Anti-spike protein IgG distribution against SARS-CoV-2 and HCoVs IgG by cohort and COVID-19 status before vaccination. Individual distribution of total concentration of anti-S IgG for SARS-CoV-2 (dark green lines), OC43 (olive green lines), HKU1 (orange lines), 229E (brown lines), NL63 (golden lines) measured in the three studied groups after the first vaccine dose. Panel (**A**) shows subjects with no history of COVID-19 infection (*n* = 39) while Panel (**B**) displays those who reported a prior natural infection (*n* = 31). Day 0 corresponds to the first vaccination. Arrows show the mean days for the second (36.6 ± 34.1 days) and booster (154.3 ± 41.1 days) doses. Individual data points were pooled and fitted with spline curves for each cohort, and gray bands indicate 95% confidence intervals.

**Figure 5 vaccines-13-00965-f005:**
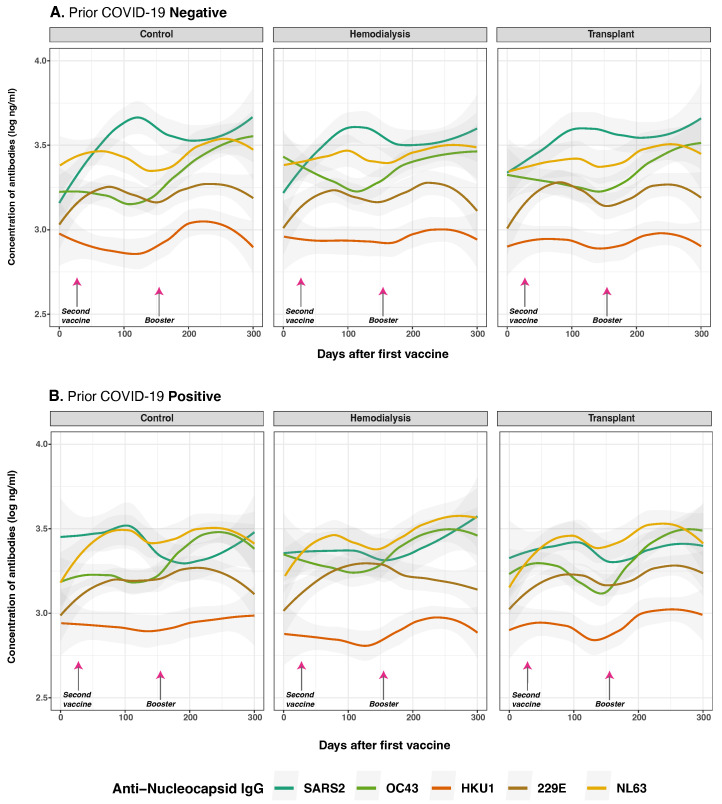
Anti-nucleocapsid protein IgG distribution against SARS-CoV-2 and HCoVs IgG by cohort and COVID-19 status before vaccination. Individual distribution of total concentration of anti-N IgG for SARS-CoV-2 (dark green lines), OC43 (olive green lines), HKU1 (orange lines), 229E (brown lines), NL63 (golden lines) measured in the three studied groups after first vaccine dose. Panel (**A**) shows subjects with no history of COVID-19 infection (*n* = 39) while Panel (**B**) displays those who reported a prior natural infection (*n* = 31). Day 0 corresponds to the first vaccination. Arrows show the mean days for the second (36.6 ± 34.1 days) and booster (154.3 ± 41.1 days) doses. Individual data points were pooled and fitted with spline curves for each cohort, and gray bands indicate 95% confidence intervals.

**Figure 6 vaccines-13-00965-f006:**
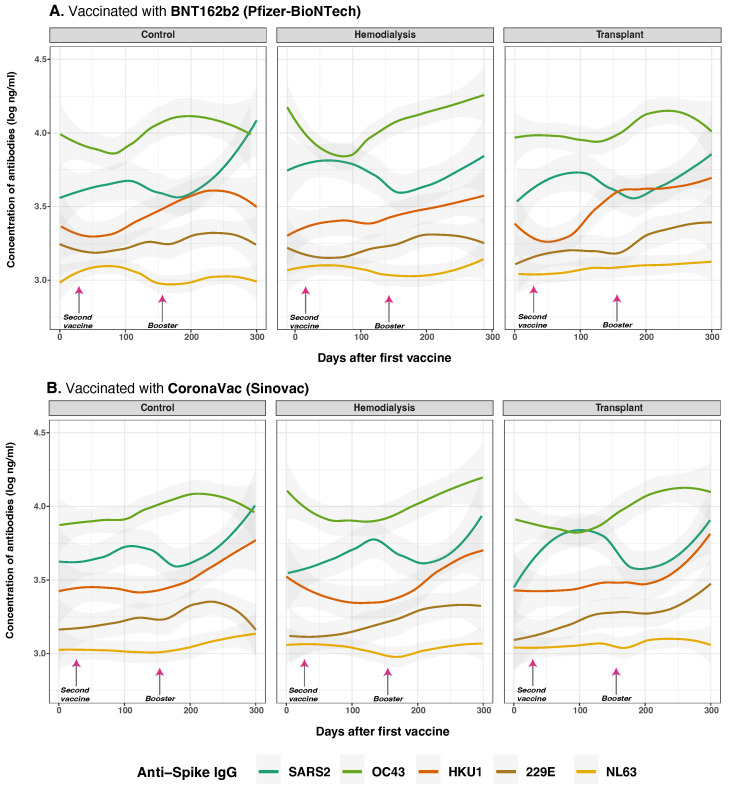
Anti-spike protein IgG distribution against SARS-CoV-2 and HCoVs IgG by cohort and vaccine type. Concentrations of total anti-S IgG for SARS-CoV-2 (dark green lines), OC43 (olive green lines), HKU1 (orange lines), 229E (brown lines), NL63 (golden lines) were measured in control (*n* = 33), hemodialysis (*n* = 13), and transplant (*n* = 24) cohorts by multiplex assay. Day 0 corresponds to the first vaccination. Arrows show the mean days for the second (36.6 ± 34.1 days) and booster (154.3 ± 41.1 days) doses. Among the subjects, 43 individuals (63.24%) were vaccinated with BNT162b2 (panel (**A**)), while 25 individuals (36.76%) received CoronaVac (panel (**B**)). Additionally, 8 subjects received homologous boosters, and seven received heterologous boosters. Individual data points were pooled and fitted with spline curves for each cohort, and gray bands indicate 95% confidence intervals.

**Figure 7 vaccines-13-00965-f007:**
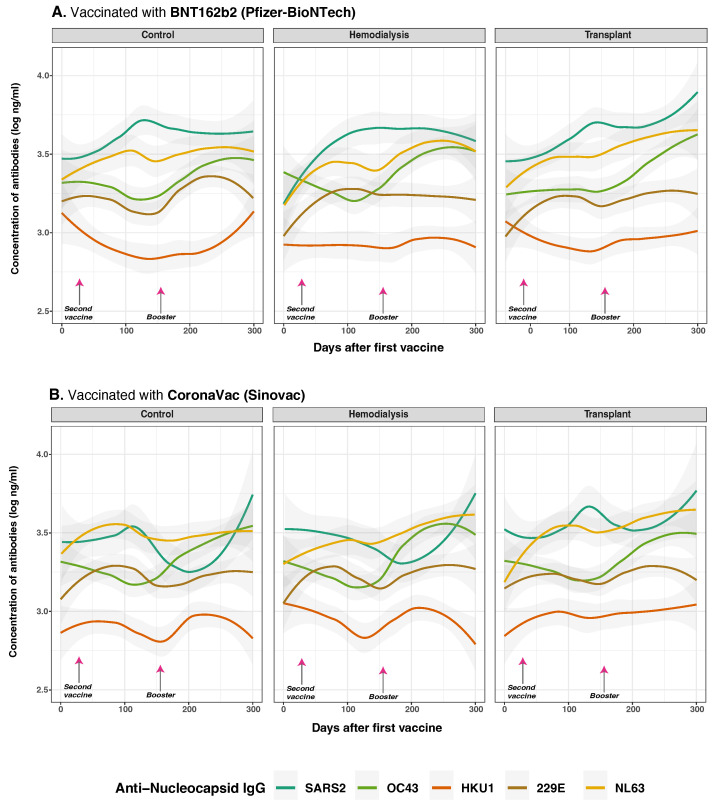
Anti-nucleocapsid protein IgG distribution against SARS-CoV-2 and HCoVs IgG by cohort and vaccine type. Concentrations of total anti-S IgG for SARS-CoV-2 (dark green lines), OC43 (olive green lines), HKU1 (orange lines), 229E (brown lines), NL63 (golden lines) were measured in control (*n* = 33), hemodialysis (*n* = 13), and transplant (*n* = 24) cohorts by multiplex assay. Day 0 corresponds to the first vaccination. Arrows show the mean days for the second (36.6 ± 34.1 days) and booster (154.3 ± 41.1 days) doses. Among the subjects, 43 individuals (63.24%) were vaccinated with BNT162b2 (panel (**A**)), while 25 individuals (36.76%) received CoronaVac (panel (**B**)). Additionally, eight subjects received homologous boosters, and seven received heterologous boosters. Individual data points were pooled and fitted with spline curves for each cohort, and gray bands indicate 95% confidence intervals.

**Table 1 vaccines-13-00965-t001:** Study Cohort: Demographics, Vaccination, and COVID-19 History. Adapted from [23]. We analyzed samples from 70 participants across three cohorts: control (*n* = 33), hemodialysis (HD; *n* = 13), and transplant (TR; *n* = 24). We stratified participants by prior COVID-19 history into positive (*n* = 31) and negative (*n* = 39) groups. In total, 271 capillary blood samples were available for analysis. Among vaccinated individuals, 43 received BNT162b2 (8 with homologous and 7 with heterologous boosters), and 25 received CoronaVac. We evaluated categorical variables using Pearson’s chi-squared and Fisher’s exact tests, and we present results as counts with percentages.

Characteristic	Control, *n* = 33 ^1^	Hemodialysis, *n* = 13 ^1^	Transplant, *n* = 24 ^1^	*p*-Value ^2^
**Age**				0.150
18–24	6 (18%)	1 (7.7%)	3 (13%)	
25–44	21 (64%)	5 (38%)	11 (46%)	
45–65	6 (18%)	7 (54%)	10 (42%)	
**Sex**				0.200
Female	19 (58%)	4 (31%)	10 (42%)	
Male	14 (42%)	9 (69%)	14 (58%)	
**Race**				<0.001
Asian	1 (3.0%)	0 (0%)	0 (0%)	
Black	15 (45%)	10 (77%)	24 (100%)	
White	7 (21%)	0 (0%)	0 (0%)	
Not reported	10 (30%)	3 (23%)	0 (0%)	
**Ethnicity**				0.009
Hispanic or Latino	19 (58%)	10 (77%)	19 (79%)	
Non-Hispanic or Non-Latino	4 (12%)	0 (0%)	5 (21%)	
Not reported	10 (30%)	3 (23%)	0 (0%)	
**Education level**				<0.001
Less than high school	1 (3.0%)	5 (38%)	1 (4.2%)	
High school graduate	4 (12%)	3 (23%)	7 (29%)	
Higher education	9 (27%)	0 (0%)	15 (63%)	
Graduate education	3 (9.1%)	0 (0%)	0 (0%)	
Not reported	16 (48%)	5 (38%)	1 (4.2%)	
**Underlying conditions**				<0.001
None	27 (82%)	0 (0%)	1 (4.2%)	
At least one	6 (18%)	2 (15%)	6 (25%)	
Two or more	0 (0%)	11 (85%)	17 (71%)	
**COVID-19 positive before first fingerstick**				0.053
Yes	10 (30%)	6 (46%)	15 (63%)	
No	23 (70%)	7 (54%)	9 (38%)	
**COVID-19 positive during the study**				>0.900
Yes	8 (24%)	2 (15%)	5 (21%)	
Not reported	25 (76%)	11 (85%)	19 (79%)	
**Vaccine doses**				<0.001
One dose	11 (33%)	5 (38%)	2 (8.3%)	
Two doses	20 (61%)	4 (31%)	11 (46%)	
Fully vaccinated and one booster	2 (6.1%)	2 (15%)	10 (42%)	
Two or more boosters	0 (0%)	0 (0%)	1 (4.2%)	
Not reported	0 (0%)	2 (15%)	0 (0%)	
**Vaccine type (prime schedule)**				0.066
BNT162b2	25 (76%)	7 (64%)	11 (46%)	
CoronaVac	8 (24%)	4 (36%)	13 (54%)	
**Booster schedule**				0.700
Heterologous	0 (0%)	1 (50%)	6 (55%)	
Homologous	2 (100%)	1 (50%)	5 (45%)	

^1^ n (%); ^2^ Fisher’s exact test; Pearson’s Chi-squared test.

## Data Availability

The original data presented in this study, along with the R code used for analysis, are openly available on FigShare. The R code is accessible at DOI: 10.60593/ur.d.29425199, and the dataset is available at DOI: 10.60593/ur.d.29425202.

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
