# Peer review of "Antibody Responses to SARS-CoV-2 and Common HCoVs in Hemodialysis Patients and Transplant Recipients: Data from the Dominican Republic"

_vaccines, 2025, doi:10.3390/vaccines13090965_

Round 1
Reviewer 1 Report
Comments and Suggestions for Authors
This manuscript is an extended analysis based on a previously published study: “Antibody Response to SARS-CoV-2 Vaccines in Transplant Recipients and Hemodialysis Patients: Data from the Dominican Republic” (Vaccines 2024, 12, 1312). The current manuscript expands the original work by comparing antibody responses to SARS-CoV-2 with responses to common human coronaviruses (HCoVs). This is a well-conducted study that addresses an important research question of antibodies against SARS-CoV-2 and HCOVs among haemodialysis and transplant recipients. The findings indicate limited but measurable cross-reactivity, which is scientifically valuable, especially in immunocompromised populations.
Comments.
1. Immunosuppressants in transplant recipients:
Given that one of the study’s focal groups is transplant recipients (TR). The absence of information on immunosuppressive regimens is a notable gap. Immunosuppressants, such as mTOR inhibitors, calcineurin inhibitors, or corticosteroids are significantly affect humoral responses.
Consider clarifying a brief discussion on the potential effects of immunosuppressive therapy on antibody responses in the Discussion section. If available, I strongly recommend including data on immunosuppressive regimens (e.g. types, doses, durations) in Table 1 or supplementary material.
2. Chronic kidney disease (CKD) staging:
The manuscript would benefit from more detailed clinical information on the haemodialysis (HD) group. The CKD stage and dialysis frequency (e.g. hours per week) are known to influence immune competence and antibody response.
Suggest clarifying the CKD stages of participants and dialysis parameters in the HD cohort, either in the Methods or Table 1.
3. Lack of neutralising antibody data:
This study measures total and anti-S IgG and anti-N IgG (binding assays), but does not include neutralising antibody titres (blocking assays), which are functionally important for viral entry blockade. In immunocompromised patients, neutralising activity may or may not correlate directly with binding IgG levels, as known in healthy individuals.
Suggest acknowledging this limitation explicitly in the Discussion or Limitations section.
4. Reliance on self-reported data:
The use of self-reported histories for prior SARS-CoV-2 infection and vaccination status is prone to recall bias and misclassification, especially given the possibility of asymptomatic infections.
Consider discussing how this limitation might affect group stratification and antibody response interpretation.
5. Ambiguity in the origin of anti-N responses:
Since CoronaVac is a whole virion inactivated vaccine. It can elicit anti-N IgG responses similar to natural infection, albeit generally at lower levels. This complicates the use of anti-N as a marker of prior infection.
Suggest clarifying how anti-N responses were interpreted in terms of infection vs. vaccination. Have you considered using anti-N kinetics (e.g. timing or decline patterns) to differentiate their origin?
6. Future research directions:
The conclusion could be strengthened by offering clearer guidance for future research. Suggesting acknowledging a larger sample size and longitudinal studies incorporating neutralising antibody assays, cellular immunity (T-cell responses), and broader geographic and ethnic diversity would provide a clear roadmap for building upon this research.
Typos.
1. Line 79: “caplillary”. Should be corrected to “capillary”.
Author Response
This manuscript is an extended analysis based on a previously published study: “Antibody Response to SARS-CoV-2 Vaccines in Transplant Recipients and Hemodialysis Patients: Data from the Dominican Republic” (Vaccines 2024, 12, 1312). The current manuscript expands the original work by comparing antibody responses to SARS-CoV-2 with responses to common human coronaviruses (HCoVs). This is a well-conducted study that addresses an important research question of antibodies against SARS-CoV-2 and HCOVs among haemodialysis and transplant recipients. The findings indicate limited but measurable cross-reactivity, which is scientifically valuable, especially in immunocompromised populations.
Comments 1. Immunosuppressants in transplant recipients: Given that one of the study’s focal groups is transplant recipients (TR). The absence of information on immunosuppressive regimens is a notable gap. Immunosuppressants, such as mTOR inhibitors, calcineurin inhibitors, or corticosteroids are significantly affect humoral responses.
Consider clarifying a brief discussion on the potential effects of immunosuppressive therapy on antibody responses in the Discussion section. If available, I strongly recommend including data on immunosuppressive regimens (e.g. types, doses, durations) in Table 1 or supplementary material.
Response 1. We thank the reviewer for this helpful suggestion. In response, we have added Supplementary Tables S1–S3 to provide further details on the study cohorts. Supplementary Table S1 summarizes the hemodialysis and transplant groups, including causes of end-stage kidney disease and immunization history, while Supplementary Table S3 provides detailed information on the transplant cohort and immunosuppressive regimens. In addition, we have included a brief paragraph in the discussion section (Lines 445–452) to address these points.
Comments 2. Chronic kidney disease (CKD) staging: The manuscript would benefit from more detailed clinical information on the haemodialysis (HD) group. The CKD stage and dialysis frequency (e.g. hours per week) are known to influence immune competence and antibody response.
Suggest clarifying the CKD stages of participants and dialysis parameters in the HD cohort, either in the Methods or Table 1.
Response 2. We thank the reviewer for this suggestion. In response, we have added Supplementary Tables S1–S2 to provide additional cohort details. Table S1 summarizes the hemodialysis and transplant cohorts, including causes of end-stage kidney disease and immunization history. Table S2 presents additional hemodialysis characteristics, such as access type and treatment duration.
Comments 3. Lack of neutralising antibody data: This study measures total and anti-S IgG and anti-N IgG (binding assays), but does not include neutralising antibody titres (blocking assays), which are functionally important for viral entry blockade. In immunocompromised patients, neutralising activity may or may not correlate directly with binding IgG levels, as known in healthy individuals.
Suggest acknowledging this limitation explicitly in the Discussion or Limitations section.
Response 3. We thank the reviewer for highlighting this point. We acknowledge that, based on correlations reported in Paper 1, the study can only infer neutralization indirectly through the association between antibodies targeting the full-length S protein and the RBD portion. However, while direct binding data are lacking, we have previously shown a high correlation between multiplex results, especially RBD binding, and viral neutralization for SARS-CoV-2 (Cameron, et al. doi:10.1128/jcm.02489-20). This current manuscript focuses specifically on describing antibody kinetics post-vaccination, comparing responses between vaccines and cohorts, and evaluating cross-reactive responses.
We also wish to point out that, in our first paper, we examined antibody concentrations against the SARS-CoV-2 full S protein and its subunits (S1, S2, RBD) individually, as well as the correlations between these levels before and after administration of either the BNT162b2 or CoronaVac vaccines. Results showed a strong association between anti-S IgG and anti-RBD IgG (R = 0.92), as well as a robust correlation with anti-S2 IgG levels (R = 0.86). While anti-RBD IgG levels strongly correlate with viral neutralization, anti-S2 IgG antibodies have been reported to associate with disease severity and may serve as markers of early immune responses. Additionally, IgG concentrations against the S-RBD and S2 subunits were also significantly correlated (R = 0.81). This prior work was not the focus of the current manuscript.
Comments 4. Reliance on self-reported data: The use of self-reported histories for prior SARS-CoV-2 infection and vaccination status is prone to recall bias and misclassification, especially given the possibility of asymptomatic infections.
Consider discussing how this limitation might affect group stratification and antibody response interpretation.
Response 4. We thank the reviewer for this important comment. Regarding COVID-19 history, as described in the Methods, all participants underwent PCR testing at enrollment, and those testing positive were excluded (line 131). Participants who reported prior or study-period COVID-19 infection were required to provide PCR confirmation and the date of diagnosis. Pre-vaccination serology was not available; therefore, the absence of symptoms or clinical signs was the only documented data.
We acknowledge that the lack of baseline serology and reliance on self-reporting represent limitations that could affect group stratification and interpretation of antibody responses. These are common issues in real-world observational studies. However, as part of our initial data exploration and validation, we performed individual analyses to determine cut-off values and assess associations with detectable antibodies at baseline, which supported the stratification of participants based on prior SARS-CoV-2 infection.
With respect to vaccination status, we note that the SARS-CoV-2 vaccines were newly introduced at the time of the study, and group prioritization was strictly regulated by the national healthcare system. Because our study cohorts were among the prioritized groups, the main enrollment challenge was the identification of unvaccinated individuals, as vaccination reports were carefully scrutinized. Therefore, the risk of misclassification, even with self-reported data, was minimal.
Comments 5. Ambiguity in the origin of anti-N responses: Since CoronaVac is a whole virion inactivated vaccine. It can elicit anti-N IgG responses similar to natural infection, albeit generally at lower levels. This complicates the use of anti-N as a marker of prior infection.
Suggest clarifying how anti-N responses were interpreted in terms of infection vs. vaccination. Have you considered using anti-N kinetics (e.g. timing or decline patterns) to differentiate their origin?
Response 5. We appreciate the comment and have included a discussion of anti-N levels in CoronaVac recipients. In addition, we address vaccine differences and their potential impact on antibody responses, particularly CoronaVac and anti-N, due to the nature of the vaccine, in multiple sections of the manuscript, including the Results (Figure 7) and the Discussion (lines 460–480). We also have added a clarification on the limitations of using anti-N as a marker of prior infection and the implications of CoronaVac for selecting booster types and schedules.
Additionally, we provided a detailed analysis of anti-N kinetics, including timing and decline patterns, in our first paper. While we touch on this briefly here, differentiating anti-N IgG responses elicited by infection, from antibodies elicited by vaccination, is not the focus of the current manuscript. We may consider exploring this in more detail in future studies.
Comments 6. Future research directions: The conclusion could be strengthened by offering clearer guidance for future research. Suggesting acknowledging a larger sample size and longitudinal studies incorporating neutralising antibody assays, cellular immunity (T-cell responses), and broader geographic and ethnic diversity would provide a clear roadmap for building upon this research
Response 6. We have modified the discussion section to highlight future directions for research (lines 507-509). One focus of the study was that there was limited data on populations outside of the United States, Europe, and Asia (lines 61-78). We hope that this work will serve as an example of how such studies can be conducted in under-resourced areas, and how international collaboration can provide local insights.
Typos. Line 79: “caplillary”. Should be corrected to “capillary”.
Response. All typos have been corrected in the revised manuscript.
Reviewer 2 Report
Comments and Suggestions for Authors
The authors examined the antibody cross-reactivities against virus antigens delivered from SARS, α-coronavirus strains (229E, NL63), and β-coronavirus strains (OC43, HKU1). The antibodies were collected from healthy controls, patients with haemodialysis, and transplanted recipients. SARS-CoV-2, a β-coronavirus, shared epitopes of the spike proteins and nucleocapsid proteins with other β-coronaviruses (OC43 and HKU1). The anti-virus antibody levels increased after SARS-CoV-2 vaccinations more clearly in healthy controls compared to those in immunocompromised patients. The authors introduced the methods of the multi-level mixed modelling (MLMM) to identify key immunological and clinical co-factors.
To conclude, more samples need to be examined.
As the authors commented, a small number of samples is the limitation of this study. It is difficult to draw clear conclusions. The authors demonstrated the methods that could be useful for further analysis of antibody activities.
Minor points:
- For supplemental information, the duration of dialysis, the time periods after transplantation, and also the plasma concentration of calcineurin inhibitors or other immunosuppressants could be good to add.
- Typos: Figure 1, haemodialysis n=15 should be n=13, line 79, caplillary should be capillary
Author Response
The authors examined the antibody cross-reactivities against virus antigens delivered from SARS, α-coronavirus strains (229E, NL63), and β-coronavirus strains (OC43, HKU1). The antibodies were collected from healthy controls, patients with haemodialysis, and transplanted recipients. SARS-CoV-2, a β-coronavirus, shared epitopes of the spike proteins and nucleocapsid proteins with other β-coronaviruses (OC43 and HKU1). The anti-virus antibody levels increased after SARS-CoV-2 vaccinations more clearly in healthy controls compared to those in immunocompromised patients.
The authors introduced the methods of the multi-level mixed modelling (MLMM) to identify key immunological and clinical co-factors.
To conclude, more samples need to be examined. As the authors commented, a small number of samples is the limitation of this study. It is difficult to draw clear conclusions. The authors demonstrated the methods that could be useful for further analysis of antibody activities.
Minor points:
Comments 1: For supplemental information, the duration of dialysis, the time periods after transplantation, and also the plasma concentration of calcineurin inhibitors or other immunosuppressants could be good to add.
Response 1. We thank the reviewer for the suggestion. In response, we have added Supplementary Tables S1–S3 to provide additional cohort details. Table S1 summarizes the hemodialysis and transplant cohorts, including causes of end-stage kidney disease and immunization history. Table S2 presents additional hemodialysis characteristics, such as access type and treatment duration. Table S3 details the transplant cohort and immunosuppression characteristics.
Typos: Figure 1, haemodialysis n=15 should be n=13, line 79, caplillary should be capillary
Response. We thank the reviewer for pointing this out. All typos have been corrected in the revised manuscript.
Reviewer 3 Report
Comments and Suggestions for Authors
This study is of interest as it examines the antibody response to vaccines for the primary prevention of COVID-19 in patients with immunocompromise, either iatrogenic, due to transplantation, or resulting from haemodialysis. The authors conducted a longitudinal analysis and assessed cross-reactivity with common human coronaviruses. There is good control of multicollinearity through MLMM and MDS; however, several limitations remain:
- The total sample comprises only 70 subjects, from which two very small subgroups emerge: 13 haemodialysis patients and 24 transplant recipients, with imbalance between them.
- COVID-19 history is self-reported and not definitively diagnosed without pre-vaccination serological confirmation.
- Short follow-up period.
Some suggestions:
- It is not reported which immunosuppressive therapy the transplant patients were receiving (type, dose).
- It is not reported how many years had elapsed since transplantation or initiation of haemodialysis.
- Anti-N antibodies are present following vaccination with CoronaVac, but it is unclear whether they can be distinguished from those due to natural infection and, for this reason, a sensitivity analysis excluding subjects with prior infection would be warranted.
- The abstract would benefit from greater detail (e.g., number of patients included) to convey the magnitude of the results.
- Given the very small patient number, this could be presented as a pilot/proof-of-concept study and this should be specified in the title, abstract, and methods, and the discussion and introduction should better clarify whether larger studies have already been published and what this study adds in comparison.
- The introduction should also cite data on the safety of this vaccine in chronic immunosuppression, including in other chronic conditions (e.g., https://pubmed.ncbi.nlm.nih.gov/36047032/). Additional data from studies in kidney disease would also be beneficial regarding safety.
Author Response
This study is of interest as it examines the antibody response to vaccines for the primary prevention of COVID-19 in patients with immunocompromise, either iatrogenic, due to transplantation, or resulting from haemodialysis. The authors conducted a longitudinal analysis and assessed cross-reactivity with common human coronaviruses. There is good control of multicollinearity through MLMM and MDS; however, several limitations remain:
- The total sample comprises only 70 subjects, from which two very small subgroups emerge: 13 haemodialysis patients and 24 transplant recipients, with imbalance between them.
- COVID-19 history is self-reported and not definitively diagnosed without pre-vaccination serological confirmation.
- Short follow-up period.
Response. We thank the reviewer for this important comment. As described in the Methods, all participants underwent PCR testing at enrollment, and those testing positive were excluded (line 131). Participants who reported prior or study-period COVID-19 infection were required to provide PCR confirmation and the date of diagnosis. Pre-vaccination serology was not available; therefore, the absence of symptoms or clinical signs was the only documented data.
We acknowledge that the lack of baseline serology and reliance on self-reporting represent limitations that could affect group stratification and interpretation of antibody responses. However, as part of our initial data exploration and validation, we performed individual analyses to determine cut-off values and assess associations with detectable antibodies at baseline, supporting stratification based on prior SARS-CoV-2 infection.
Regarding vaccination status, this vaccine was newly introduced at the time of the study, and group prioritization was strictly regulated. Our cohorts were among the prioritized populations, so the main enrollment challenge was identifying unvaccinated individuals. Vaccination records were carefully verified, minimizing the risk of misclassification, even with self-reported data.
Some suggestions:
Comments 1. It is not reported which immunosuppressive therapy the transplant patients were receiving (type, dose).
Response 1. We thank the reviewer for the suggestion. In response, we have added Supplementary Tables S1 and S3 to provide additional cohort details. Table S1 summarizes the hemodialysis and transplant cohorts, including causes of end-stage kidney disease and immunization history. Table S3 details the transplant cohort and immunosuppression characteristics. (Lines 140-141).
Comments 2. It is not reported how many years had elapsed since transplantation or initiation of haemodialysis.
Response 2. We appreciate the reviewer’s suggestion. Additional cohort details have been added to the Supplementary Materials, with time since dialysis initiation and transplantation now included in Supplementary Tables S1 and S3.
Comments 3. Anti-N antibodies are present following vaccination with CoronaVac, but it is unclear whether they can be distinguished from those due to natural infection and, for this reason, a sensitivity analysis excluding subjects with prior infection would be warranted.
Response 3. We appreciate the observation. We discuss the impact of vaccine type on antibody responses, particularly CoronaVac and anti-N, in the Results (Figure 7) and Discussion (Lines 460–480), including limitations of using anti-N as a marker of prior infection and implications for booster selection. While anti-N kinetics were addressed in our prior study, differentiating responses from infection versus vaccination is not the focus of this manuscript and may be explored in future work.
Comments 4. The abstract would benefit from greater detail (e.g., number of patients included) to convey the magnitude of the results.
Response 4. Thank you for this helpful suggestion. We have revised the abstract to include the number of subjects in each cohort (CO: n = 33; HD: n = 13; TR: n = 24) to better convey the magnitude of the results.
Comments 5. Given the very small patient number, this could be presented as a pilot/proof-of-concept study and this should be specified in the title, abstract, and methods, and the discussion and introduction should better clarify whether larger studies have already been published and what this study adds in comparison.
Response 5. We thank the reviewer for this thoughtful suggestion. The current manuscript acknowledges the sample size limitation. We also note that while larger studies have examined antibody reactivity to SARS-CoV-2, no prior studies have looked at the correlation with antibodies directed against the other four circulating coronaviruses in dialysis patients and renal transplant recipients, and none that we are aware of have comparison data for CoronaVac. This study adds these specific aspects, and the introduction has been modified to note this.
Respectfully, the current title, abstract, and methods remain the same. We already discuss study limitations, and reference other studies in the introduction and discussion.
Comments 6. The introduction should also cite data on the safety of this vaccine in chronic immunosuppression, including in other chronic conditions (e.g., https://pubmed.ncbi.nlm.nih.gov/36047032/). Additional data from studies in kidney disease would also be beneficial regarding safety. BNT162b2 mRNA COVID-19 vaccine is safe in a setting of patients on biologic therapy with inflammatory bowel diseases: a monocentric real-life study
Response 6. Respectfully, a discussion of vaccine safety is beyond the scope of this study. Our manuscript briefly addresses the association between immunosuppression and vaccine response in individuals with renal disease on hemodialysis, and renal transplant recipients. A detailed discussion of vaccine safety in the wide variety of immunosuppressed populations, including patients with other chronic conditions or autoimmune disease, falls outside the scope of our analysis. This study was not designed to evaluate safety differences between vaccine platforms.
Round 2
Reviewer 1 Report
Comments and Suggestions for Authors
Thank you for thoroughly addressing the concerns. The changes made have improved the clarity and overall quality of the manuscript.
Reviewer 2 Report
Comments and Suggestions for Authors
The authors responded well. I have nothing to add.
Reviewer 3 Report
Comments and Suggestions for Authors
The authors made the requested revisions. No other comments.